# Examining the predictors of use of sanitary napkins among adolescent girls: A multi-level approach

Shekhar Chauhan[1], Pradeep Kumar[2], Strong Pillar Marbaniang[2], Shobhit Srivastava[2], Ratna Patel[3]*, Preeti Dhillon[4]

1 Department of Population Policies and Programmes, International Institute for Population Sciences, Mumbai, India, 2 Department of Mathematical Demography & Statistics, International Institute for Population Sciences, Mumbai, India, 3 Department of Public Health and Mortality Studies, International Institute for Population Sciences, Mumbai, India, 4 Department of Mathematical Demography & Statistics, International Institute for Population Sciences, Mumbai, India

* ratnapatelbhu@gmail.com

## Abstract

### Background

This paper aimed to explore various factors associated with the use of sanitary napkins among adolescent girls in Uttar Pradesh and Bihar.

### Methods

The study uses information from the Understanding the Lives of Adolescents and Young Adults (UDAYA) project survey conducted in Uttar Pradesh and Bihar in 2016. The study sample consisted of 14,625 adolescent girls aged 10–19 years. The study sample was selected using a multi-stage systematic sampling design. Multilevel logistic regression (MLR) was used to identify the individual and community level factors associated with the use of sanitary napkins.

### Results

The results revealed a wide variation in sanitary napkins' use across the socio-economic and demographic factors. The use of sanitary napkins was significantly higher among girls with 8–9 (53.2%) and 10 and more (75.4%) years of schooling compared to those who had no formal education (26.4%). The use of sanitary napkins was higher among adolescent girls who were not engaged in paid work (54.7%) than those who did any paid job (40.8%). Adolescent girls reporting frequent exposure to mass media (OR = 2.10), belonging to the richest wealth quintile (OR = 3.76), and whose mothers had 10 or more years of education (OR = 2.29) had a higher propensity to use sanitary napkins than their counterparts. We did not find a significant role of community-level education of mothers on the menstrual hygiene practices of adolescents.

**Data Availability Statement:** https://dataverse.harvard.edu/dataset.xhtml?persistentId=doi:10.7910/DVN/RRXQNT.

**Funding:** This paper was written using data collected as part of Population Council's UDAYA study, which is funded by the Bill and Melinda Gates Foundation and the David and Lucile Packard Foundation. No additional funds were received for the preparation of the paper.

**Competing interests:** The authors have declared that no competing interests exist.

## Conclusion

Ensuring that adolescent girls have access to hygienic means to manage their menses is critical from a public health perspective and in enabling them to realize their full potential. Programs to enhance menstrual hygiene are warranted. These programs should involve mothers, who are an important source of knowledge about menstrual hygiene. Facilitating girls' access to education may also produce tangible menstrual hygiene benefits.

## Introduction

Menstruation is a normal cycle and a healthy part of girls' and women's lives, but there are some cultural and religious misconceptions regarding the menstrual period [1]. Hygienic menstrual management consists of using a clean menstrual management material to absorb or collect blood that can be changed in privacy as frequently as necessary for the duration of the menstruation period, use of soap and water to wash the body as required, and availability of facilities to dispose of used menstrual management materials [2]. The benefits of maintaining good hygiene during periods include a reduced risk of urinary tract infections, genitals rashes, and cervical cancer [3–5]. Inadequate menstrual hygiene management among adolescent girls (15–19 years) is a public health concern, particularly in low and middle-income countries [6]. India has 355 million menstruating girls and women. Millions of these women and girls face a significant barrier to a comfortable and dignified experience with menstrual hygiene management [7].

To promote menstrual hygiene practices among rural adolescent girls, the Indian government in 2011 implemented an initiative to increase awareness on menstrual hygiene, access to and use of high-quality sanitary napkins, and safe disposal of sanitary napkins in an environmentally friendly manner [8]. However, a study in 2005 showed that 90% of India's women used an unhygienic cloth to manage their menstrual period, whereas only 11.2% used hygienic sanitary pads and 3.9% used locally prepared napkins [3]. A qualitative study in rural Haryana in 2006 revealed that only 30% of adolescent girls used sanitary napkins, even though 80% of them were aware of them and 79% were motivated to use sanitary napkins [9].

Poor menstrual hygiene practices due to limited accessibility to water and sanitation, lack of privacy, and unaffordability of sanitary pads can result in adverse health outcomes like reproductive tract infections (RTIs) [1,10] and increased absenteeism among adolescent school girls [11]. Untreated RTIs contribute to 10–15% of fetal wastage and 30–50% of prenatal infection [1]. Reproductive tract infections are also associated with the incidence of cervical cancer, HIV/AIDS, infertility, ectopic, pregnancy, and a myriad of other symptoms [1]. Qualitative studies report that the fear and humiliation from leakage of blood and body odour result in menstruating girls being absent from school [12].

Gopalan (2019) points out three barriers to adopting menstrual hygiene practices in India: the lack of awareness about menstruation, a lack of acceptance about the menstrual period, and lack of access to quality hygiene products [13]. Reports from a meta-analysis of 138 studies showed that about half of adolescent girls in India were not aware of menarche's causes when it started, with only a quarter understanding the source of bleeding [12]. A large study in India revealed that 70% of women cited cost as the main barrier for not using sanitary pads [11]. Other reasons for not using sanitary napkins were problems with disposal of used sanitary napkins [9]. Apart from individual-level factors, menstrual hygiene practices such as disposal

of used menstrual products and discussing menstrual hygiene are also influenced by some community characteristics such as cultural taboos [14].

This paper aimed to explore the factors associated with the use of sanitary napkins among adolescent girls in Uttar Pradesh and Bihar. Existing studies on menstrual hygiene practices among adolescent girls in India are based on micro-level data from different pockets of the country and failed to consider the impact of community characteristics on the use of sanitary napkins [9,11]. Community-level characteristics do not merely indicate the distribution of resources and opportunities in a population; they drive interesting and important social dynamics, which cannot be captured using individual characteristics alone [14]. Hence, this study's contribution is to establish a relationship between the individual and community level characteristics and the use of sanitary napkins among adolescent girls.

## Methods

### Data

The study utilized data from the Understanding the Lives of Adolescents and Young Adults (UDAYA) project survey conducted in two Indian states, "Uttar Pradesh and Bihar", in 2016 by Population Council under the guidance of the Ministry of Health and Family Welfare, Government of India. The survey collected detailed information on family, media, community environment, assets acquired in adolescence, and quality of transitions to young adulthood indicators. Uttar Pradesh and Bihar's sample size was 10,350 and 10,350 adolescents aged 10–19 years, respectively. UDAYA was designed to provide estimates for the state as a whole as well as for the urban and rural areas of the state for each of the five categories of respondents, namely younger boys of 10–14 years, older boys of 15–19 years, younger girls of 10–14 years, unmarried older girls of 15–19 years, and married older girls in ages 15–19. The required sample for each sub-group of adolescents was determined at 920 younger boys, 2,350 older boys, 630 younger girls, 3,750 older girls, and 2,700 married girls in both states.

The study treated the state's rural and urban areas as independent sampling domains and, therefore, drew sample areas independently for each of these two domains. The 150 primary sampling units (PSUs) were further divided equally into rural and urban areas, 75 for rural respondents and 75 for urban respondents. Within each sampling domain, the study adopted a multi-stage systematic sampling design. The 2011 census list of villages and wards (each consisting of several census enumeration blocks [CEBs] of 100–200 households) served as the sampling frame for selecting villages and wards in rural and urban areas, respectively. This list was stratified using four variables: region, village/ward size, the proportion of the population belonging to scheduled castes and scheduled tribes, and female literacy.

The household sample in rural areas was selected in three stages, while in urban areas, it was selected in four stages. In rural areas, villages were first selected systematically from the stratified list described above, with selection probability proportional to size (PPS). In urban areas, 75 wards were first selected systematically with probability proportional to size. CEBs were then arranged by their administrative number within each selected ward, and one CEB was randomly selected. Several CEBs adjacent to the selected CEB were merged to ensure at least 500 households for listing. The details about sampling design and survey methodology are published elsewhere [15,16]. Permission was granted to access the dataset for the analysis purpose.

The effective sample size for this study was 14,016 menstruating adolescents girls aged 10–19 years. About 609 girls (4.2%) were excluded from the sample because they had not started menstruating.

### Variable description

**Outcome variable.** The outcome variable was binary in nature, defining the use of sanitary napkins coded as 1 "yes" if the girl was using sanitary napkins or 0 "no." The variable was generated using the question "Girls use different things during the menstrual period to prevent blood stains from becoming evident. What do you use for this?" The responses were a. locally prepared napkins b. use sanitary napkins c. use cloth d. used nothing, e. use other (specify). Responses were recoded as 1 (locally prepared napkins/sanitary napkins) and 0 (use cloth/ used nothing/use other (specify). Additionally, in some cases, girls were using both napkins and cloths; those cases were included as use of sanitary napkins only (less than 1% cases). Locally prepared napkins are the napkins prepared indigenously and are much cheaper than the regular sanitary napkins widely available in the market [17].

**Explanatory variables.** This study's explanatory variables were taken after considering previously available literature [3,18,19]. Age was coded as 10–12, 13–14,15–17, 18–19 years. Education was coded as "no education," "1–7 years", "8–9 years," and "10 or more years". The educational status was defined for the respondent's education. Working status was coded as "no" and "yes." Working status was defined as the respondents who did paid work in the last one year. Media exposure assessed the extent to which the respondent was exposed to television, radio, or newspapers. The media exposure variable was coded as "no," "rare," and "frequent". Mother's education was coded as "no education," "1–7 years", "8–9 years," and "10 or more years".

Wealth index was coded as "poorest," "poorer," "middle," "richer," and "richest." Households are given scores based on the number and kinds of consumer goods they own, ranging from a television to a bicycle or car, and housing characteristics such as source of drinking water, toilet facilities, and flooring materials. These scores are derived using principal component analysis. National wealth quintiles are compiled by assigning the household score to each usual (*de jure*) household member, ranking each person in the household population by their score, and then dividing the distribution into five equal categories, each with 20 percent of the population.

Caste was coded as "Scheduled Caste/Scheduled Tribe (SC/ST)" and "non-SC/ST." The Scheduled Caste includes "untouchables" that is socially and financially/economically segregated by their low status. The Scheduled Castes (SCs) and Scheduled Tribes (STs) are among the most disadvantaged socio-economic groups in India [20]. Religion was coded as "Hindu" and "non-Hindu." Residence was available in the data as "urban" and "rural." Survey was conducted in two states, "Uttar Pradesh" and "Bihar."

Community-level variables were constructed by aggregating individual/household-level characteristics of the respondents at the primary sampling unit (PSU) level. The UDAYA data provided a household wealth index (WI) based on information collected on household amenities and assets. The community economic index was divided into two categories, low and high, with low being for PSUs whose average household WI was less than the national average of WI and high being that for the remaining PSUs [21]. Similarly, the individual's educational index was created based on the average years of schooling of women at the PSU level [21]. A similar index for mother's education was also created. A community media exposure index was also created based on average media exposure at the PSU level and then dividing it into low and high as per average media exposure. India's government has stressed the importance of mass-media exposure in creating awareness regarding sanitary napkins usage [22].

**Statistical analysis.** We used bivariate analysis (chi-square tests) to examine the association between the outcome variable and other socio-demographic predictors. We employed multilevel logistic regression to assess the effect of the individual-, household (family)-, and

community-level variables on the use of sanitary napkin among adolescent girls. The random effects of household and community were estimated using *melogit* command in STATA (version 15).

The multilevel modeling application is justified by the hierarchal structure of the survey, where adolescents were nested within households, and the households were nested within PSUs. We first ran a null model, that is, without keeping any predictors. The null model represented the total variance in the use of sanitary napkins at household and community levels. We then fitted three models; in the first model, we included individual-level predictors. The second model included individual and household level variables. In the final model (Model 3), we added community-level variables in addition to individual and household level predictors. The significance of random effects was evaluated for all the estimated models by using p-values at a 95% confidence interval.

The mathematical description of the final model (three levels) is given below:

$$logit\left(\pi_{ijk}\right) = log\left(\frac{\pi_{ijk}}{1 - \pi_{ijk}}\right) = \beta_{0jk} + \beta_1 x_{1ijk} + \beta_2 x_{2ijk} + \beta_3 x_{3ijk} + \cdots + \beta_n x_{nijk}$$

Here, $\pi_{ijk} = p(y_{ijk} = 1)$ is the probability that adolescents (i) in the household j, from the PSU k, use a sanitary napkin. Where $y_{ijk}$ is equal to "1" if an adolescent girl uses a sanitary napkin and "0" if she did not. The study defined this probability as a function of an intercept and the exploratory variables as follows: $\beta_{0jk} = \beta_0 + \mu_{0jk}$.

In this equation, $\beta_{0jk}$ indicates that the paper modeled the intercept in this relationship as random at $j^{th}$ (household) and $k^{th}$ (PSU) levels. The variables $x_{1ijk}$ to $x_{nijk}$ were the exploratory variables, and their coefficients were fixed effects. The technical advantage of this methodology relies on the error term structure. Linear or logistic regression models exhibit one error term for the whole equation, whereas multilevel analysis generates one error term for each level, isolating the individual-level and group-level residual variance. The split error term in the multilevel analysis allows assessing unobserved effects at every level [21].

## Results

The socio-demographic profile of adolescent girls is presented in *Table 1*. The majority of girls were aged 15–19 years (92.2 per cent), about one-third of adolescents had 10 or more years of schooling, and only 17.1 per cent were working. Around half of the adolescents reported frequent exposure to mass media, and three-fourths of girls' mothers had no schooling. One-fourth of girls belonged to the scheduled caste/scheduled tribe group, 78.5 per cent were Hindu, and about 84 per cent lived in rural areas.

The use of sanitary napkins among adolescent girls by background characteristics are presented in Table 2. The use of sanitary napkins was significantly higher among late adolescents (54.5 per cent), and it was low among early adolescent girls (26.1 per cent). A higher proportion of girls with 8–9 (53.2 per cent) and 10 or more (75.4 per cent) years of schooling used sanitary napkins than those who were uneducated (26.4 per cent). The use of sanitary napkins was higher among adolescents who were not working (54.7 per cent) than those working (40.8 per cent). Moreover, sanitary napkins were more prevalent among adolescents who reported frequent exposure to mass media (65.8 percent), and it was lowest among those who had no exposure to media (25.6 percent). Mother's education and wealth index were significantly positively associated with the use of sanitary napkins. A higher proportion of girls whose mothers had 10 or more years of education (84.3 percent) used sanitary napkins than those whose mothers had no education (46 percent). The proportion of girls who used sanitary napkins was higher among girls from the richest households than those from poorer households.

**Table 1. Socio-demographic profile of adolescent girls.**

| Variables | Sample | Percentage |
|---|---|---|
| **Age (years)** | | |
| Early adolescents (10–14) | 1,096 | 7.8 |
| Late adolescents (15–19) | 12,920 | 92.2 |
| **Education (years)** | | |
| No education | 1,869 | 13.3 |
| 1–7 | 3,437 | 24.5 |
| 8–9 | 4,000 | 28.5 |
| 10 or more | 4,710 | 33.6 |
| **Working status** | | |
| No | 11,613 | 82.9 |
| Yes | 2,403 | 17.1 |
| **Media exposure** | | |
| No exposure | 2,605 | 18.6 |
| Rarely | 4,031 | 28.8 |
| Frequently | 7,380 | 52.7 |
| **Mother's education (years)** | | |
| No education | 10,501 | 74.9 |
| 1–7 | 1,382 | 9.9 |
| 8–9 | 958 | 6.8 |
| 10 or more | 1,176 | 8.4 |
| **Wealth index** | | |
| Poorest | 1,868 | 13.3 |
| Poorer | 2,616 | 18.7 |
| Middle | 3,033 | 21.6 |
| Richer | 3,430 | 24.5 |
| Richest | 3,068 | 21.9 |
| **Caste** | | |
| SC/ST | 3,629 | 25.9 |
| Non-SC/ST | 10,387 | 74.1 |
| **Religion** | | |
| Hindu | 11,003 | 78.5 |
| Non-Hindu | 3,013 | 21.5 |
| **Residence** | | |
| Urban | 2,273 | 16.2 |
| Rural | 11,743 | 83.8 |
| **State** | | |
| Uttar Pradesh | 9,435 | 67.3 |
| Bihar | 4,581 | 32.7 |
| **Total** | 14,016 | 100.0 |

SC/ST: Scheduled caste/Scheduled tribe.

Moreover, the use of sanitary napkins was lower among SC/ST (45.6 per cent) and non-Hindu (48.8 per cent) adolescents compared to non-SC/ST (54.7 per cent) and Hindu adolescents (53.3 per cent). The use of sanitary napkins was significantly higher among girls who lived in urban areas than those who lived in rural.

**Table 2. Percentage distribution of adolescent girls who use sanitary napkins by background characteristics.**

| Variables | Percentage | P<0.05 |
|---|---|---|
| **Age (years)** | | * |
| Early adolescents (10–14) | 26.1 | |
| Late adolescents (15–19) | 54.5 | |
| **Education (years)** | | * |
| No education | 26.4 | |
| 1–7 | 33.9 | |
| 8–9 | 53.2 | |
| 10 and above | 75.4 | |
| **Working status** | | * |
| No | 54.7 | |
| Yes | 40.8 | |
| **Media exposure** | | * |
| No exposure | 25.6 | |
| Rarely | 44.9 | |
| Frequently | 65.8 | |
| **Mother's education (years)** | | * |
| No education | 46.0 | |
| 1–7 | 61.9 | |
| 8–9 | 69.1 | |
| 10 and above | 84.3 | |
| **Wealth index** | | * |
| Poorest | 29.1 | |
| Poorer | 34.4 | |
| Middle | 47.6 | |
| Richer | 58.8 | |
| Richest | 79.3 | |
| **Caste** | | * |
| SC/ST | 45.6 | |
| Non-SC/ST | 54.7 | |
| **Religion** | | |
| Hindu | 53.3 | |
| Non-Hindu | 48.8 | |
| **Residence** | | * |
| Urban | 71.1 | |
| Rural | 48.7 | |
| **State** | | |
| Uttar Pradesh | 54.0 | |
| Bihar | 48.8 | |
| **Total** | 52.3 | |

SC/ST: Scheduled caste/Scheduled tribe

*if p<0.05.

The estimates from multilevel logistic regression analysis, showing the odds ratios with 95% confidence interval of factors associated with the use of the sanitary napkin among girls, are presented in *Table 3*. Model 1 included individual-level predictor variables. Model 2 included household-level variables in addition to the explanatory variables used in Model 1, and Model

**Table 3. Multilevel logistic regression analysis assessing the effect of background characteristics on the likelihood of use of sanitary napkins among girls.**

| Variables | Model-1 | Model-2 | Model-3 |
|---|---|---|---|
| | OR (95%CI) | OR (95%CI) | OR (95%CI) |
| **Age (years)** | | | |
| Early adolescents (10–14) | Ref. | Ref. | Ref. |
| Late adolescents (15–19) | 3.57*(2.85,4.46) | 3.85*(3.06,4.86) | 3.85*(3.06,4.86) |
| **Education (years)** | | | |
| No education | Ref. | Ref. | Ref. |
| 1–7 | 1.51*(1.27,1.79) | 1.46*(1.23,1.73) | 1.48*(1.25,1.76) |
| 8–9 | 3.42*(2.81,4.17) | 3.05*(2.52,3.71) | 3.12*(2.57,3.78) |
| 10 and above | 8.17*(6.33,10.55) | 6.45*(5.05,8.25) | 6.65*(5.19,8.51) |
| **Working status** | | | |
| No | Ref. | Ref. | Ref. |
| Yes | 0.66*(0.57,0.77) | 0.76*(0.66,0.88) | 0.78*(0.67,0.90) |
| **Media exposure** | | | |
| No exposure | Ref. | Ref. | Ref. |
| Rarely | 1.57*(1.34,1.85) | 1.49*(1.27,1.76) | 1.47*(1.25,1.73) |
| Frequently | 3.33*(2.75,4.02) | 2.45*(2.05,2.94) | 2.10*(1.74,2.53) |
| **Mother's education (years)** | | | |
| No education | Ref. | Ref. | Ref. |
| 1–7 | 1.62*(1.35,1.93) | 1.43*(1.20,1.71) | 1.41*(1.18,1.68) |
| 8–9 | 2.01*(1.61,2.50) | 1.69*(1.36,2.10) | 1.66*(1.34,2.06) |
| 10 and above | 3.35*(2.65,4.24) | 2.43*(1.93,3.06) | 2.29*(1.82,2.87) |
| **Wealth index** | | | |
| Poorest | | Ref. | Ref. |
| Poorer | | 1.09(0.90,1.31) | 1.08(0.89,1.30) |
| Middle | | 1.64*(1.35,1.98) | 1.56*(1.29,1.88) |
| Richer | | 2.01*(1.64,2.47) | 1.82*(1.49,2.23) |
| Richest | | 4.33*(3.34,5.62) | 3.76*(2.92,4.85) |
| **Caste** | | | |
| SC/ST | | Ref. | Ref. |
| Non-SC/ST | | 1.25*(1.10,1.43) | 1.23*(1.08,1.41) |
| **Religion** | | | |
| Hindu | | Ref. | Ref. |
| Non-Hindu | | 0.93(0.79,1.09) | 0.90(0.77,1.05) |
| **Community wealth index** | | | |
| Low | | | Ref. |
| High | | | 1.45*(1.15,1.84) |
| **Community education index** | | | |
| Low | | | Ref. |
| High | | | 1.14*(1.02,1.39) |
| **Community media index** | | | |
| Low | | | Ref. |
| High | | | 1.23*(1.02,1.48) |
| **Community education (mother)** | | | |
| Low | | | Ref. |
| High | | | 1.03(0.73,1.45) |
| **Residence** | | | |
| Urban | | | Ref. |

(*Continued*)

**Table 3.** (Continued)

| Variables | Model-1 | Model-2 | Model-3 |
|---|---|---|---|
| | OR (95%CI) | OR (95%CI) | OR (95%CI) |
| Rural | | | 0.59*(0.47,0.74) |
| **State** | | | |
| Uttar Pradesh | | | Ref. |
| Bihar | | | 1.40*(1.16,1.69) |

SC/ST: Scheduled caste/Scheduled tribe

*if p<0.05

OR: Odds Ratio; CI: Confidence Interval.

3 added community-level variables. In Model 1, age, adolescents' educational level, working status, mass media exposure, and mother's education status were significantly associated with the use of sanitary napkins. Model 3 revealed that the use of sanitary napkins was 3.85 times significantly higher among late adolescent girls (OR: 3.85; CI: 3.06–4.86) than early adolescents. The likelihood of sanitary napkin use was 1.48, 3.12, and 6.65 times higher among girls with 1–7 (OR: 1.48; CI: 1.25–1.76), 8–9 (OR: 3.12; CI: 2.57–3.78), and 10 or more (OR: 6.65; CI: 5.19–8.51) years of schooling compared to adolescents with no education. The odds of sanitary napkin use were 22 percent lower among working girls (OR: 0.78; CI: 0.67–0.90) than those who were not working. Moreover, girls who reported rare (OR: 1.47; CI: 1.25–1.73) or frequent (OR: 2.10; CI: 1.74–2.53) exposure to mass media was significantly more likely to use sanitary napkins compared to those who had no exposure to mass media. Mother's education has a significant effect on the use of sanitary napkins among girls.

The likelihood of sanitary napkin use was significantly higher among girls from the middle (OR: 1.56; CI: 1.29–1.88), richer (OR: 1.82; CI: 1.49–2.23), and richest (OR: 3.76; CI: 2.92–4.85) households compared to poorest ones. Compared to SC/ST girls, non-SC/ST girls had higher odds of sanitary napkin use (OR: 1.23; CI: 1.08–1.41).

Girls from communities with a higher wealth index (OR: 1.45; CI: 1.15–1.84) had higher odds of sanitary napkin use than girls from communities with a low wealth index. Similarly, girls from communities with a higher education index (OR: 1.14; CI: 1.02–1.39) had higher odds of sanitary napkin use than girls from communities with low education levels. Girls from communities with high media exposure (OR: 1.23; CI: 1.02–1.48) were 23 per cent more likely to use sanitary napkins than those from communities with low media exposure. Moreover, girls from rural areas were 41 per cent less likely (OR: 0.59; CI: 0.47–0.74) to use sanitary napkins than those from urban areas.

A null model (without covariates) for the use of sanitary napkins among adolescent girls (_Table 4_) revealed a significant amount of variation across households and communities.

**Table 4. Variance estimates across families and communities, and intra-class correlation coefficient for the multilevel models for the use of sanitary napkin among adolescent girls.**

| Random Effect Parameters | Null | Model 1 | Model 2 | Model 3 |
|---|---|---|---|---|
| Community (PSU) random variance (SE) | 1.38 (0.18) | 0.67 (0.09) | 0.55 (0.08) | 0.36 (0.05) |
| Household random variance (SE) | 1.57 (0.38) | 1.27 (0.36) | 1.30 (0.37) | 1.27 (0.37) |
| Community (PSU) ICC (%) | 0.22 | 0.13 | 0.11 | 0.07 |
| Household ICC (%) | 0.47 | 0.37 | 0.36 | 0.33 |

SE: Standard error; PSU: Primary sampling unit; ICC: Intra-class correlation coefficient.

Based on intra-class correlation coefficient (ICC) values, 47 per cent and 22 per cent of the total variance in the use of sanitary napkins among girls were attributable to differences across families and communities, respectively. After including individual- (Model 1), household- (Model 2), and community-level variables (Model 3) in the null model, the ICC values decreased to 7 per cent (community level) and 33 per cent (household level). The results suggest that the likelihood of sanitary napkin use was influenced by a similar decision of another girl from the same household and/or community.

## Discussion

Even though this study focused on adolescent girls only, the study adds relevant information to the available literature. In this study, we examined the predictors of sanitary napkin use among adolescent girls by adopting a multi-level approach. For a comprehensive analysis, we examined four important community-level variables, namely, community wealth index, community education index, community media index, and community education index (for mothers), along with individual and household level variables. We identified several predictors of the use of sanitary napkins among adolescent girls.

Adolescents aged 15–19 years were more likely to use sanitary napkins than those aged 10–14 years. Older adolescents tend to have high levels of education, which may explain their higher use of sanitary napkins. We found that adolescents with higher education were more likely to use sanitary napkins than those with no education. Other researchers have acknowledged the importance of education in promoting sanitary use among adolescent girls [18]. Education among adolescent girls promotes their mass-media exposure, which further promotes sanitary use [23]. The relationship between sanitary napkins and education is bi-direction as the use of sanitary napkins may also promote girls' education [24]. Montgomery et al., in their study, noted that the distribution of sanitary napkins improves school attendance among girls [24]. The results from multi-level analysis further noted that a high level of community education is associated with higher use of sanitary napkins among adolescent girls.

Media exposure among adolescent girls is one of the important predictors of sanitary use. We found that adolescents with exposure to media were more likely to use sanitary napkins than adolescents who had no exposure to media. Similar findings have been reported elsewhere [25–27]. Mass media is a major source of information about menstrual hygiene; therefore, girls with frequent exposure to mass media may tend to use more sanitary napkins than their counterparts [28]. The use of mass media communication can be a useful tool for the development of knowledge on menstruation [19].

Higher levels of mother's education were associated with higher use of sanitary napkins among adolescent girls. This finding is in agreement with previously available literature [26]. However, the association between community education (mother) and the use of sanitary napkins was not significant. Mothers play a crucial role in educating their daughters about health matters [10,26]. Having an educated mother can therefore play a significant role in maintaining menstrual hygiene [28,29]. Furthermore, the importance of mothers' education was outlined by the fact that girls whose mothers were uneducated were more likely to miss school during their menstruation than girls whose mothers were educated [30]. Since mothers are girls' primary source of information on menstruation; mother education plays an important role in maintaining menstrual hygiene [31].

The wealth index of the household is another predictor of the use of sanitary napkins among adolescents. The study found that a higher wealth index was associated with higher use of sanitary napkins among adolescent girls. A previous study also concluded that the use of sanitary napkins was higher among adolescents in rich households than among adolescents in poor

households [32]. Adolescents are dependent on their parents to buy them sanitary napkins; thus, poor parents may find it difficult to purchase sanitary napkins for their daughters [33,34].

We found that the use of sanitary napkins was higher among non-SC/ST adolescents than their counterparts. In the Indian context, SC/ST tend to have poorer socio-economic status, limiting the use of sanitary napkins among them [35]. Furthermore, we found that the use of sanitary napkins was higher among adolescent girls in urban areas than in rural areas. Similar findings have been reported elsewhere [36,37]. Higher use of sanitary napkins among urban girls may be attributed to their higher education level, better school infrastructure, and easy access to sanitary napkins. In India, sanitary pads are sold in chemists shop and departmental stores, which are more common in urban areas [38]. Narayan et al. suggested that urban girls have a higher level of awareness about hygienic menstrual practices than their rural counterparts [39]. Moreover, girls may avoid buying sanitary napkins from shops with male shopkeepers in rural areas due to shame [32]. Furthermore, disposing of sanitary napkins is another issue in the rural area, and therefore rural women find cloth a comfortable medium to use during menstruation [40].

## Limitations of the study

Despite providing in-depth information about the predictors of sanitary napkins among adolescent girls, the study has some limitations. The study was conducted only in two Indian states, namely, Uttar Pradesh and Bihar; therefore, the findings are not generalizable to the country population. The data are cross-sectional and therefore limits our understanding of causal inferences.

## Conclusion

Ensuring that adolescent girls have access to hygienic means to manage their menses is critical from a public health perspective and in enabling them to realize their full potential. Although India's Prime Minister Shri Narendra Modi in his 74th Independence Day speech on 15th August 2020, stressed the importance of sanitary napkins, current national programs do not emphasize menstrual hygiene. Programs to enhance menstrual hygiene are warranted. The study could not find a significant community-level education effect on menstrual hygiene but revealed a strong effect of mother's as well as, individual's education and exposure to mass media on menstrual hygiene practices. Therefore, we recommend individual-level awareness programs on menstrual hygiene are needed. These programs should also involve mothers, who are an important source of knowledge about menstrual hygiene. Facilitating girls' access to education may also produce tangible menstrual hygiene benefits.

## Acknowledgments

This research uses data from the study on "Understanding the Lives of Adolescents and Young Adults (UDAYA) in Bihar and Uttar Pradesh," collected by the Population Council. Therefore, the authors are thankful to the Population Council for providing the data. The authors are also thankful to David Jean Simon for copyediting the manuscript.

## Author Contributions

**Conceptualization:** Pradeep Kumar, Shobhit Srivastava, Ratna Patel.

**Data curation:** Shobhit Srivastava.

**Formal analysis:** Pradeep Kumar, Shobhit Srivastava.

**Investigation:** Pradeep Kumar, Shobhit Srivastava.

**Methodology:** Pradeep Kumar, Shobhit Srivastava.

**Software:** Pradeep Kumar, Shobhit Srivastava.

**Supervision:** Shobhit Srivastava, Ratna Patel, Preeti Dhillon.

**Validation:** Pradeep Kumar, Shobhit Srivastava, Ratna Patel.

**Visualization:** Shobhit Srivastava.

**Writing – original draft:** Shekhar Chauhan, Strong Pillar Marbaniang, Ratna Patel.

**Writing – review & editing:** Ratna Patel, Preeti Dhillon.

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
