## [Decision Letter · Decision Letter 0]

11 Jan 2021

PONE-D-20-32727

Examining the predictors of use of sanitary napkins among adolescent girls: A multi-level approach

PLOS ONE

Dear Dr. Patel,

Thank you for submitting your manuscript to PLOS ONE. After careful consideration, we feel that it has merit but does not fully meet PLOS ONE’s publication criteria as it currently stands. Therefore, we invite you to submit a revised version of the manuscript that addresses the points raised during the review process.

Considering the reviewers' report and my own reading of this paper, I feel the paper has merit for publication in PLOS One. However, the paper needs a major revision before it can be recommending for publication. I have the following suggestion to make this paper more strong in terms of conceptual and theoretical clarity on the subject. My comments are as follows

The introduction and background can be improved. The contribution of the paper needs to be strengthened. Authors write “the pioneering contribution of this study is to establish a relationship between the individual and community level characteristics and the use of sanitary napkins among adolescent girls”. I don’t think running a multi-level logistic regression makes a paper “pioneering contribution”. Just aggregating individual and household level characteristics to a sampling cluster that most often spreads across the communities and villages are not novel at all. Yes, you can use it to model a multi-level logistic regression as it satisfies your statistical criteria. Moreover, for an outcome variable like “Sanitary Napkin use” the community doesn’t fall in physical space in a world of virtual space with the entry of digital media. Probably, this is one reason why only community media coming as a significant variable among all community variables considered for the study. Support the design for construction of the community variable with the previous literature or add a few lines about the limitation of the community variable constructed for the study using its reliability and validity statistics.

Currently, the explanatory variables are written very badly, can you write them in a paragraph mode with a little more explanation and selection justification.

The following paper in the Uttar Pradesh context might help you to tighten your paper introduction and discussion.

MALHOTRA A, GOLI S, COATES S, MOSQUERA-VASQUEZ MA. Factors associated with knowledge, attitudes, and hygiene practices during menstruation among adolescent girls in Uttar Pradesh. Waterlines. 2016 Jul 1:277-305.

 https://www.jstor.org/stable/26600766?seq=1

Carefully revise the paper according to other reviewers' comments.

We look forward to receiving your revised manuscript.

Kind regards,

Srinivas Goli, Ph.D.

Academic Editor

PLOS ONE

Additional Editor Comments:

Considering the reviewers' report and my own reading of this paper, I feel the paper has merit for publication in PLOS One. However, the paper needs a major revision before it can be recommending for publication. I have the following suggestion to make this paper more strong in terms of conceptual and theoretical clarity on the subject. My comments are as follows

1. The introduction and background can be improved. The contribution of the paper needs to be strengthened. Authors write “the pioneering contribution of this study is to establish a relationship between the individual and community level characteristics and the use of sanitary napkins among adolescent girls”. I don’t think running a multi-level logistic regression makes a paper “pioneering contribution”. Just aggregating individual and household level characteristics to a sampling cluster that most often spreads across the communities and villages are not novel at all. Yes, you can use it to model a multi-level logistic regression as it satisfies your statistical criteria. Moreover, for an outcome variable like “Sanitary Napkin use” the community doesn’t fall in physical space in a world of virtual space with the entry of digital media. Probably, this is one reason why only community media coming as a significant variable among all community variables considered for the study. Support the design for construction of the community variable with the previous literature or add a few lines about the limitation of the community variable constructed for the study using its reliability and validity statistics.

2. Currently, the explanatory variables are written very badly, can you write them in a paragraph mode with a little more explanation and selection justification.

3. The following paper in the Uttar Pradesh context might help you to tighten your paper introduction and discussion.

MALHOTRA A, GOLI S, COATES S, MOSQUERA-VASQUEZ MA. Factors associated with knowledge, attitudes, and hygiene practices during menstruation among adolescent girls in Uttar Pradesh. Waterlines. 2016 Jul 1:277-305.

https://www.jstor.org/stable/26600766?seq=1

4. Carefully revise the paper according to other reviewers' comments.

Journal Requirements:

2. In the Methods section of your manuscript, please address the following:

-    Please provide further clarification on how media exposure was stratified.

-    Please clarify within the manuscript text whether permission was granted prior to access to the dataset.

Reviewers' comments:

Reviewer's Responses to Questions

**Comments to the Author**

1. Is the manuscript technically sound, and do the data support the conclusions?

Reviewer #1: Yes

Reviewer #2: Partly

Reviewer #3: Yes

Reviewer #4: Yes

2. Has the statistical analysis been performed appropriately and rigorously? 

Reviewer #1: Yes

Reviewer #2: Yes

Reviewer #3: Yes

Reviewer #4: Yes

3. Have the authors made all data underlying the findings in their manuscript fully available?

Reviewer #1: Yes

Reviewer #2: No

Reviewer #3: No

Reviewer #4: Yes

4. Is the manuscript presented in an intelligible fashion and written in standard English?

Reviewer #1: Yes

Reviewer #2: No

Reviewer #3: Yes

Reviewer #4: Yes

5. Review Comments to the Author

Reviewer #1: Sanitary napkin use is a important predictor of adolescent health and well being..Few comments:

1.Research question is novel and important

2. The authors use the word multilevel approach but didn’t explain it clearly in the methods section

3. Statistical analysis has been performed well

4. Discussion should be written in light of the major findings particularly the findings which are contradictory to literature

Reviewer #2: In this manuscript, the authors examine the correlates of sanitary napkin use among adolescents in Uttar Pradesh and Bihar in India. Given the need to ensure that menstruators are able to manage their menses in a safe, hygienic and dignified way, the study addresses an important issue. That said, there are several limitations to the current work. These are summarized below, however, additional comments are made in the manuscript in tracked changes

• The manuscript needs a close read through for grammar. I have made extensive editorial suggestions in the attached Word document that I hope the authors will find useful.

• Given that the authors use cross-sectional data; they should be careful not to use language that implies causality. This aspect (the inability to make causal inferences) could also be highlighted as a limitation

• The authors should clarify whether their data is restricted to menstruating girls. If so, it would be useful to know what proportion of girls had experienced menarche in the larger dataset. If not, then the analysis should be limited to those who had begun menstruating.

• Given that the journal audience is international, the authors should explain terms that may not be widely understood (e.g., locally prepared napkins, scheduled caste, etc.)

• The discussion would benefit from a more nuanced discussion of the implications of the findings. Importantly, while sanitary napkins are important in managing menstruation, they are not the only product that enables menstruators to manage their menses in a hygienic manner.

• The authors note that acknowledgements are not applicable. However, the UDAYA data use requirements, indicate that authors should cite the UDAYA, Adolescent Survey as the data source in all reports, presentations, and publications based on the data with the following suggested acknowledgment: This research uses data from the study on "Understanding the Lives of Adolescents and Young Adults (UDAYA) in Bihar and Uttar Pradesh" which was collected by the Population Council.

Reviewer #3: Summary- The article explores the various factors determining the use of sanitary napkins among adolescent girls in Uttar Pradesh and Bihar. The study finds that education, exposure to mass-media and economic status significantly influence the use of sanitary napkins among adolescent girls

Merits - The study uses a large dataset. The conclusions are mostly well supported by the results. It is also quite well written, apart from lack of clarity in some places. I see no major flaws except for minor revisions as below

Page 4 Methods

•What was the basis of sample size calculation?

•Define subgroups and their basis of sampling

•Define stages of multistage systematic sampling

Page 5 & 6 Explanatory variables

•Define working status

•Give reference for household and community wealth index calculation

•What was the basis of categorization of community media exposure? (Reference)

Page 10 Results-Table 3

•Define low vs high community media exposure and give possible explanation for lower sanitary napkins use in communities with high media exposure

•Give possible explanation for low sanitary napkin use among non-working

•Explanation for lesser use of sanitary napkins in communities with high wealth index

Page 13 Discussion

•Possible explanations for lower use of sanitary napkins among rural girls could be its availability, affordability or could be some perceptions they hold about the use of sanitary napkins, but the explanation given with reference no. 31 seems to be a weird one.

Reviewer #4: Title: Examining the predictors of use of sanitary napkins among adolescent girls: A multilevel approach

Review comments to the author

Summary of the research

Authors have tried to establish linkages between individual, household and community level factors on the use of sanitary napkins by young girls. The research topic is interesting and of high issue for girls of lower- and middle-income countries. Overall, the study is well designed and written in lucid and clear language, however writing quality could have improved to some extent to bring more clarity. I observed that somewhere the authors have used tenses casually or given incomplete sentences/recommendation making the text unclear to readers.

Abstract:

1. The authors state that ‘Involve different stakeholder…….sanitary napkins’. The statement doesn’t clarify, who should involve stakeholders – program implementing partner or government or NGOs?? It should be stated clearly for whom the recommendations are made for.

Introduction:

1. The first paragraph of the introduction section mentioned that ‘According to Sarma, (2018), ….. unitary tract infections,….. cervical cancer. Authors should write carefully about the infections quoted from other studies. Here, ‘…unitary tract infection…’ has no meaning and diverting the relevance of the topic.

2. Authors should introduce abbreviations before using it repeatedly. For eg. In the second paragraph of introduction, RTI is used several time without being mention what does it stands for.

3. In the sentence ‘Increasingly RTIs….ectopic, pregnancy …..symptoms.’; ectopic pregnancy are used together without the use of comma ( , )

Figures and tables:

1. Authors have used different styles for writing Table titles: for eg. Table-1, Table-2, Table-3, Table 4: , somewhere title indent is in middle, somewhere it is in left. One should use uniform style as preferred by the journal.

2. Figure in tables should use uniform number of decimals, some places figure in tables/ confidence intervals used whole number, somewhere it is in 2 decimals. Please make them uniform

Methods:

1. In the data section, the author has written ‘The required sample….in both states.’ The author however has not explained the age group to differentiate the younger boys/girls from older boys/girls. Also, the sample of 2700 married girls are not clear if these married girls’ sample are included in younger/older age group or excluded and made separate category irrespective of the younger/older category.

2. Authors can add a section on limitations of the research, if any.

Results:

1. Author has casually used the tenses in result section. For eg: while describing Table 3 author has written ‘Model 1 includes individual-level…..’ and further Model 2 included household-level…’ ; ‘Mother’s education has a significant……’. Use of tenses should be used appropriately.

Discussions:

1. The sentence is not clear ‘The use of sanitary napkin …….. reproductive tract infections among’. This sentence is incomplete as a reader cannot understand who the audience are? Among whom?

2. Author has casually written community education index twice in the sentence ‘For a comprehensive analysis….household level variables’.

Conclusion:

1. Author has written ‘Furthermore, there is a need to provide sanitary napkin at an affordable price on a large scale, specifically in rural areas’. Authors should be careful while writing such incomplete recommendation. This statement is not clear to me, who should provide sanitary napkins? What is the rationale behind who should provide sanitary napkins?

.

6. PLOS authors have the option to publish the peer review history of their article (what does this mean?). If published, this will include your full peer review and any attached files.

Reviewer #1: **Yes: **Dr. MD Abu Bashar

Reviewer #2: No

Reviewer #3: No

Reviewer #4: **Yes: **Supriya Verma

---

## [Author Response · Author response to Decision Letter 0]

3 Feb 2021

Title: Examining the predictors of use of sanitary napkins among adolescent girls: A multilevel approach

Comments related to Journal’s requirement:

 Response: The revised manuscript is following all the requirements suggested by the journal.

2. In the Methods section of your manuscript, please address the following:

- Please provide further clarification on how media exposure was stratified.

Response: comment incorporated 

- Please clarify within the manuscript text whether permission was granted prior to access to the dataset.

Response: Comment incorporated. 

Editor’s comments:

1. The introduction and background can be improved. The contribution of the paper needs to be strengthened. Authors write “the pioneering contribution of this study is to establish a relationship between the individual and community level characteristics and the use of sanitary napkins among adolescent girls”. I don’t think running a multi-level logistic regression makes a paper “pioneering contribution”. Just aggregating individual and household level characteristics to a sampling cluster that most often spreads across the communities and villages are not novel at all. Yes, you can use it to model a multi-level logistic regression as it satisfies your statistical criteria. Moreover, for an outcome variable like “Sanitary Napkin use” the community doesn’t fall in physical space in a world of virtual space with the entry of digital media. Probably, this is one reason why only community media coming as a significant variable among all community variables considered for the study. Support the design for construction of the community variable with the previous literature or add a few lines about the limitation of the community variable constructed for the study using its reliability and validity statistics.

Response: The introduction as well as discussion section has been improved on the suggested lines. Furthermore, other suggestions provided in the comments were incorporated during the revision. We have revised the methodology and accordingly there is change in the study findings too.

2. Currently, the explanatory variables are written very badly, can you write them in a paragraph mode with a little more explanation and selection justification.

Response: Comment incorporated. 

3. The following paper in the Uttar Pradesh context might help you to tighten your paper introduction and discussion.

MALHOTRA A, GOLI S, COATES S, MOSQUERA-VASQUEZ MA. Factors associated with knowledge, attitudes, and hygiene practices during menstruation among adolescent girls in Uttar Pradesh. Waterlines. 2016 Jul 1:277-305.

https://www.jstor.org/stable/26600766?seq=1

Response: Authors are thankful to the reviewer for suggesting a very relatable piece of work. It indeed helped in improving the manuscript. Simultaneously, the same has been included as a reference.

4. Carefully revise the paper according to other reviewers' comments.

Response: Authors have carefully revised the paper as per given suggestions.

Reviewer #1: 

Sanitary napkin use is an important predictor of adolescent health and wellbeing. Few comments:

1.Research question is novel and important.

Response: Thank You for acknowledging the importance of research question.

2. The authors use the word multilevel approach but didn’t explain it clearly in the methods section.

Response: The multilevel approach has been explained clearly in the last paragraph of page number 6 and also before results section.

3. Statistical analysis has been performed well.

Response: Thank you for acknowledgment.

4. Discussion should be written in light of the major findings particularly the findings which are contradictory to literature.

Response: Authors have improved the discussion as suggested by the reviewer.

Reviewer #2: 

In this manuscript, the authors examine the correlates of sanitary napkin use among adolescents in Uttar Pradesh and Bihar in India. Given the need to ensure that menstruators are able to manage their menses in a safe, hygienic and dignified way, the study addresses an important issue. That said, there are several limitations to the current work. These are summarized below, however, additional comments are made in the manuscript in tracked changes.

Response: Authors are thankful to the reviewer for providing his/her comments through additional file. These comments and suggestions helped immensely in improving the overall quality of the paper. Authors have included the suggestions as suggested by the reviewer through additional file.

• The manuscript needs a close read through for grammar. I have made extensive editorial suggestions in the attached Word document that I hope the authors will find useful.

Response: Authors have improved the manuscript as suggested by the reviewer through attachment. Moreover, authors took help from one of the native English speaker to improve the manuscript.

• Given that the authors use cross-sectional data; they should be careful not to use language that implies causality. This aspect (the inability to make causal inferences) could also be highlighted as a limitation.

Response: A limitation (related to cross-sectional data and causality inference) has been added as suggested by the reviewer. Furthermore, authors interpreted the result keeping the issue in mind. 

• The authors should clarify whether their data is restricted to menstruating girls. If so, it would be useful to know what proportion of girls had experienced menarche in the larger dataset. If not, then the analysis should be limited to those who had begun menstruating.

Response: Yes, data is restricted to menstruating girls. About 609 girls (4.2%) were excluded from the sample as they did not started menstruating.

• Given that the journal audience is international, the authors should explain terms that may not be widely understood (e.g., locally prepared napkins, scheduled caste, etc.).

Response: Comments incorporated as suggested by the reviewer. Authors have defined the required terms elaborately.

• The discussion would benefit from a more nuanced discussion of the implications of the findings. Importantly, while sanitary napkins are important in managing menstruation, they are not the only product that enables menstruators to manage their menses in a hygienic manner.

Response: Authors have edited the discussion as suggested by the reviewer. Authors have made required changes and also added relevant information in the discussion section.

• The authors note that acknowledgements are not applicable. However, the UDAYA data use requirements, indicate that authors should cite the UDAYA, Adolescent Survey as the data source in all reports, presentations, and publications based on the data with the following suggested acknowledgment: This research uses data from the study on "Understanding the Lives of Adolescents and Young Adults (UDAYA) in Bihar and Uttar Pradesh" which was collected by the Population Council.

Response: Authors are thankful to the reviewer for pointing out this issue. Accordingly, authors have edited the acknowledgement section as suggested.

Reviewer #3: 

Summary- The article explores the various factors determining the use of sanitary napkins among adolescent girls in Uttar Pradesh and Bihar. The study finds that education, exposure to mass-media and economic status significantly influence the use of sanitary napkins among adolescent girls.

Response: Authors are thankful to the reviewer for critically reading the article with interest.

Merits - The study uses a large dataset. The conclusions are mostly well supported by the results. It is also quite well written, apart from lack of clarity in some places. I see no major flaws except for minor revisions as below.

Response: Authors are thankful to the reviewer for praising the importance of this study.

Page 4 Methods

•What was the basis of sample size calculation?

Response: Only those adolescent girls were included who started menstruating. The effective sample size for this study was 14,016 adolescents girls aged 10-19 years. About 609 girls (4.2%) were excluded from the sample as they did not started menstruating.

•Define subgroups and their basis of sampling.

Response: Thanks for the suggestion. Amendment has been done.

•Define stages of multistage systematic sampling.

Response: Stages of multistage systematic sampling has been added in the methods section.

•Define working status.

Response: Comment incorporated. Working status has been defines accordingly.

•Give reference for household and community wealth index calculation.

Response: Reference has been added corresponding to household and community wealth index calculation.

•What was the basis of categorization of community media exposure? (Reference).

Response: A reference has been provided as suggested by the reviewer.

Page 10 Results-Table 3

•Define low vs high community media exposure and give possible explanation for lower sanitary napkins use in communities with high media exposure.

Response: After reanalyzing the data, (after removal of girls who did not started menstruating from the data set) the odds are now changed and higher sanitary napkin use was observable in girls with high media exposure. 

•Give possible explanation for low sanitary napkin use among non-working.

Response: The findings were other way round and not what the reviewer understood. The usage of sanitary napkin was not low but high among non-working adolescent girls. This, of course, calls for further studies in this domain. One probable reason may be that non-working girls may be receiving money from their parents to purchase sanitary napkin. Moreover, the absence of type of work is another limitation in this study. We only know whether the girls were working or not but do not know their occupation/income which directly affect their purchasing capacity. 

•Explanation for lesser use of sanitary napkins in communities with high wealth index.

Response: After reanalyzing the data, (after removal of girls who did not start menstruating from the data set) the odds are now changed and higher sanitary napkin use was observed among girls with high wealth index category.

Page 13 Discussion

•Possible explanations for lower use of sanitary napkins among rural girls could be its availability, affordability or could be some perceptions they hold about the use of sanitary napkins, but the explanation given with reference no. 31 seems to be a weird one.

Response: Authors are thankful to the reviewer for pointing out the issue. Authors also feel that the reference is a weird one and accordingly the same has been deleted in the revised manuscript.

Reviewer 4:

Summary of the research

Authors have tried to establish linkages between individual, household and community level factors on the use of sanitary napkins by young girls. The research topic is interesting and of high issue for girls of lower- and middle-income countries. Overall, the study is well designed and written in lucid and clear language, however writing quality could have improved to some extent to bring more clarity. I observed that somewhere the authors have used tenses casually or given incomplete sentences/recommendation making the text unclear to readers. 

Response: Authors are thankful to the reviewer for praising the overall quality of the article. Furthermore, as suggested by the reviewer, authors have improved the clarity throughout the manuscript. For this, authors have taken help from native English speakers.

Abstract: 

1. The authors state that ‘Involve different stakeholder…….sanitary napkins’. The statement doesn’t clarify, who should involve stakeholders – program implementing partner or government or NGOs?? It should be stated clearly for whom the recommendations are made for.

Response: This recommendation is for government. Government shall involve various stakeholders in developing information, education, and communication for promoting the use of sanitary napkin. Stakeholders such as NGOs working in a particular region/area tend to have vast knowledge about that area/region and also understand the prevailing cultural taboos and therefore involving them may bring a required change.

Introduction:

1. The first paragraph of the introduction section mentioned that ‘According to Sarma, (2018),…..unitary tract infections,….. cervical cancer. Authors should write carefully about the infections quoted from other studies. Here, ‘…unitary tract infection…’ has no meaning and diverting the relevance of the topic. 

Response: Comment have been incorporated

2. Authors should introduce abbreviations before using it repeatedly. For eg. In the second paragraph of introduction, RTI is used several time without being mention what does it stands for. 

Response: The abbreviation RTI (reproductive tract infection) has been introduced before using it repeatedly. 

3. In the sentence ‘Increasingly RTIs….ectopic, pregnancy …..symptoms.’; ectopic pregnancy are used together without the use of comma ( , ) 

Response: Thank you for the comment. Comma has now been used at the appropriate place.

Figures and tables:

1. Authors have used different styles for writing Table titles: for eg. Table-1, Table-2, Table-3, Table 4: , somewhere title indent is in middle, somewhere it is in left. One should use uniform style as preferred by the journal. 

Response: Now we have used uniform style following the journal’s recommendations on table submission.

2. Figure in tables should use uniform number of decimals, some places figure in tables/ confidence intervals used whole number, somewhere it is in 2 decimals. Please make them uniform.

Response: Amendment has been done as suggested by the reviewer.

Methods:

1. In the data section, the author has written ‘The required sample….in both states.’ The author however has not explained the age group to differentiate the younger boys/girls from older boys/girls. Also, the sample of 2700 married girls are not clear if these married girls’ sample are included in younger/older age group or excluded and made separate category irrespective of the younger/older category. 

Response: Now it is mention in the data and method section. The required details sought by the reviewer has been included in the method section. 

2. Authors can add a section on limitations of the research, if any.

Response: As per given suggestion, authors have added a section on limitation just before the conclusion section.

Results:

1. Author has casually used the tenses in result section. For eg: while describing Table 3 author has written ‘Model 1 includes individual-level…..’ and further Model 2 included household-level…’ ; ‘Mother’s education has a significant……’. Use of tenses should be used appropriately. 

Response: Now tenses have been taken care while writing. Authors have written result in uniform style throughout manuscript. 

Discussions: 

1. The sentence is not clear ‘The use of sanitary napkin …….. reproductive tract infections among’. This sentence is incomplete as a reader cannot understand who the audience are? Among whom? 

Response: Authors are thankful to the reviewer for pointing out the error. Accordingly, the sentence has been omitted from the revised manuscript.

2. Author has casually written community education index twice in the sentence ‘For a comprehensive analysis….household level variables’. 

Response: There are two variables related to community education index. One is measuring education of the community and another variable is measuring community education of mother. 

Conclusion:

1. Author has written ‘Furthermore, there is a need to provide sanitary napkin at an affordable price on a large scale, specifically in rural areas’. Authors should be careful while writing such incomplete recommendation. This statement is not clear to me, who should provide sanitary napkins? What is the rationale behind who should provide sanitary napkins? 

Response: Authors have removed the recommendation as it was confusing.

.

---

## [Decision Letter · Decision Letter 1]

29 Mar 2021

PONE-D-20-32727R1

Examining the predictors of use of sanitary napkins among adolescent girls: A multi-level approach

PLOS ONE

Dear Dr. Patel,

Thank you for submitting your manuscript to PLOS ONE. After careful consideration, we feel that it has merit but does not fully meet PLOS ONE’s publication criteria as it currently stands. Therefore, we invite you to submit a revised version of the manuscript that addresses the points raised during the review process.

ACADEMIC EDITOR: Considering favourable opinions from the reviewers, I am going with a decision of Minor revision before the final acceptance of this piece. 

We look forward to receiving your revised manuscript.

Kind regards,

Srinivas Goli, Ph.D.

Academic Editor

PLOS ONE

Journal Requirements:

Additional Editor Comments (if provided):

Considering favourable opinions from the reviewers, I am going with a decision of Minor revision before the final acceptance of this piece.

Reviewers' comments:

Reviewer's Responses to Questions

**Comments to the Author**

1. If the authors have adequately addressed your comments raised in a previous round of review and you feel that this manuscript is now acceptable for publication, you may indicate that here to bypass the “Comments to the Author” section, enter your conflict of interest statement in the “Confidential to Editor” section, and submit your "Accept" recommendation.

Reviewer #2: (No Response)

Reviewer #3: (No Response)

2. Is the manuscript technically sound, and do the data support the conclusions?

Reviewer #2: Yes

Reviewer #3: Yes

3. Has the statistical analysis been performed appropriately and rigorously? 

Reviewer #2: Yes

Reviewer #3: Yes

4. Have the authors made all data underlying the findings in their manuscript fully available?

Reviewer #2: (No Response)

Reviewer #3: (No Response)

5. Is the manuscript presented in an intelligible fashion and written in standard English?

Reviewer #2: Yes

Reviewer #3: (No Response)

6. Review Comments to the Author

Reviewer #2: The authors have done a good job addressing the reviewers’ comments. Below are some outstanding issues that need to be addressed

Abstract

The word “examining” in the first sentence should be replaced with “associated with”

Add a space between “(UDAYA)” and “project”

The abstract conclusion is unclear and needs to be revised. Essentially, the authors found that education, exposure to mass media and economic status were significantly associated with the use of sanitary towels and propose that “it is important to bring improvements” in these indicators. However, more reflection on these findings and their implications is needed. For example, the association between economic status and sanitary towel use, suggests that the cost of sanitary towels might be prohibitive for some girls and that policies and programs that make sanitary products more financially accessible may be warranted. In the same vein, it is unclear what the authors mean by “improve mass media exposure among girls.”

Introduction

Paragraph 1: Revise the following sentence…” The benefits of maintaining good hygiene during periods help to reduce the risk of urinary tract infections, reduce the incidence of rashes in genitals, ensure good reproductive health, and minimize the risk of cervical cancer [3,4,5].” To “The benefits of maintaining good hygiene during periods include a reduced risk of urinary tract infections, genital rashes, and cervical cancer [3,4,5].”

Paragraph 3: There is no need to capitalize ‘Reproductive Tract Infections’

Last paragraph: Revise first sentence – “This paper aimed to explore the factors associated with the use of sanitary napkins…” The framing is important – in a cross-sectional study you cannot assess the factors that DETERMINE sanitary napkin use, you can only speak of associations

Methods (data)

Paragraph 1: add a comma before “Uttar Pradesh and Bihar” in the first sentence

Paragraph 1: revise second to last sentence to remove the word “in” before “ages”. The word appears several times

Paragraph 2: the acronym PSU is not defined

Last paragraph: Correct “About 609 girls (4.2%) were excluded from the sample because they had not started menstruating.”

Last paragraph: delete the last sentence, which is repetitive given the next text additions

Methods (explanatory variables)

Revise the description of media exposure ‘Media exposure assessed the extent to which the respondent was exposed to television, radio, or newspapers. The media exposure variable was coded as “no,” “rare,” and “frequent.”’

It may be better to have a separate paragraph to explain the wealth index and to clarify that the scores were then summed up and divided into five quintiles.

Remove comma after “untouchables” and correct to “…that is socially and financially segregated by their low status…”

Results

Paragraph 1: Correct “…adolescent girls IS presented…”

Table 1, seems that the font of the sample and percentage columns is different. Similar use of different font types noted in Table 2, Table 3, and Table 4.

Paragraph 2: insert space between “higher” and “among”

Paragraph 3: insert space between “were” and “22”

Discussion

Paragraph 3: insert space between “elsewhere” and the cited references

Paragraph 6: Merge and correct sentence “In India, sanitary pads are sold in chemists and departmental stores, which are more common in urban areas [38].” Delete the next sentence

Limitations

Correct “Further, the data ARE cross-sectional and causal inferences cannot be made.”

Conclusion

The conclusion should be revised. Sanitary napkins are just one product – there are tampons, menstrual cups etc., which are considered hygienic products. I would suggest that the authors reframe this as “Ensuring that adolescent girls have access to hygienic means to manage their menses is critical from a public health perspective and in enabling them to realize their full potential. Although……do not emphasize menstrual hygiene. Programs to enhance menstrual hygiene are warranted. These program should involve mothers, who are an important source of knowledge about menstrual hygiene. Facilitating girls’ access to education may also produce tangible menstrual hygiene benefits.”

The same conclusion could also be adapted for the abstract

Reviewer #3: The basis of sample size calculation has not been explained, what was the reference and formula to calculate the sample size, explain.

7. PLOS authors have the option to publish the peer review history of their article (what does this mean?). If published, this will include your full peer review and any attached files.

Reviewer #2: No

Reviewer #3: **Yes: **Dr. Harmanjeet Kaur

---

## [Author Response · Author response to Decision Letter 1]

10 Apr 2021

Editor’s comments:

Response: The authors have reviewed the reference list. It is correct. There are 40 references in manuscript text as well as in the reference section. Also, authors have checked for the authenticity of the references. None of the papers cited in the manuscript have been retracted.

Reviewer #2: The authors have done a good job addressing the reviewers’ comments. Below are some outstanding issues that need to be addressed.

Response: The authors are thankful to the reviewer for revising the manuscript critically and for providing the comments.

Abstract

The word “examining” in the first sentence should be replaced with “associated with”

Add a space between “(UDAYA)” and “project” The abstract conclusion is unclear and needs to be revised. Essentially, the authors found that education, exposure to mass media and economic status were significantly associated with the use of sanitary towels and propose that “it is important to bring improvements” in these indicators. However, more reflection on these findings and their implications is needed. For example, the association between economic status and sanitary towel use, suggests that the cost of sanitary towels might be prohibitive for some girls and that policies and programs that make sanitary products more financially accessible may be warranted. In the same vein, it is unclear what the authors mean by “improve mass media exposure among girls.”

Response: The abstract has been amended as suggested. The conclusion section has been updated on the suggested lines. 

Introduction

Paragraph 1: Revise the following sentence…” The benefits of maintaining good hygiene during periods help to reduce the risk of urinary tract infections, reduce the incidence of rashes in genitals, ensure good reproductive health, and minimize the risk of cervical cancer [3,4,5].” To “The benefits of maintaining good hygiene during periods include a reduced risk of urinary tract infections, genital rashes, and cervical cancer [3,4,5].”

Paragraph 3: There is no need to capitalize ‘Reproductive Tract Infections’

Last paragraph: Revise first sentence – “This paper aimed to explore the factors associated with the use of sanitary napkins…” The framing is important – in a cross-sectional study you cannot assess the factors that DETERMINE sanitary napkin use, you can only speak of associations.

Response: The introduction part has been modified as per suggestions provided by the reviewer.

Methods (data)

Paragraph 1: add a comma before “Uttar Pradesh and Bihar” in the first sentence

Response: The correction has been made as suggested.

Paragraph 1: revise second to last sentence to remove the word “in” before “ages”. The word appears several times

Response: The correction has been made as suggested.

Paragraph 2: the acronym PSU is not defined Last paragraph: Correct “About 609 girls (4.2%) were excluded from the sample because they had not started menstruating.” Last paragraph: delete the last sentence, which is repetitive given the next text additions

Response: The correction has been made as suggested.

Methods (explanatory variables) Revise the description of media exposure ‘Media exposure assessed the extent to which the respondent was exposed to television, radio, or newspapers. The media exposure variable was coded as “no,” “rare,” and “frequent.”’

Response: The correction has been made as suggested.

It may be better to have a separate paragraph to explain the wealth index and to clarify that the scores were then summed up and divided into five quintiles.

Remove comma after “untouchables” and correct to “…that is socially and financially segregated by their low status…”

Response: The correction has been made as suggested.

Results

Paragraph 1: Correct “…adolescent girls IS presented…”

Response: The correction has been made as suggested.

Table 1, seems that the font of the sample and percentage columns is different. Similar use of different font types noted in Table 2, Table 3, and Table 4.

Response: The font has been corrected as suggested. Now, the same font has been used in all the tables.

Paragraph 2: insert space between “higher” and “among”

Response: The correction has been made as suggested.

Paragraph 3: insert space between “were” and “22”

Response: The correction has been made as suggested.

Discussion

Paragraph 3: insert space between “elsewhere” and the cited references

Response: The correction has been made as suggested.

Paragraph 6: Merge and correct sentence “In India, sanitary pads are sold in chemists and departmental stores, which are more common in urban areas [38].” Delete the next sentence

Response: The correction has been made as suggested.

Limitations

Correct “Further, the data ARE cross-sectional and causal inferences cannot be made.”

Response: The correction has been made as suggested.

Conclusion

The conclusion should be revised. Sanitary napkins are just one product – there are tampons, menstrual cups etc., which are considered hygienic products. I would suggest that the authors reframe this as “Ensuring that adolescent girls have access to hygienic means to manage their menses is critical from a public health perspective and in enabling them to realize their full potential. Although……do not emphasize menstrual hygiene. Programs to enhance menstrual hygiene are warranted. These program should involve mothers, who are an important source of knowledge about menstrual hygiene. Facilitating girls’ access to education may also produce tangible menstrual hygiene benefits.” The same conclusion could also be adapted for the abstract

Response: The authors are thankful to the reviewer for providing the valuable suggestions related to conclusion. Authors have accordingly updated the conclusion as suggested by the reviewer.

Reviewer #3: The basis of sample size calculation has not been explained, what was the reference and formula to calculate the sample size, explain.

Response: This study is based on secondary data conducted by Population Council and team and authors did not calculated sample size. Therefore sample size calculation and formula has not given in the manuscript. More details about sampling design and survey methodology are published elsewhere (Santhya et al., 2017) and cited in the manuscript.

1. Santhya KG, Acharya R, Pandey N, Gupta A, Rampal S, Singh S, Zavier AF. Understanding the lives of adolescents and young adults (UDAYA) in Uttar Pradesh, India. New Delhi. 2017. https://www.popcouncil.org/uploads/pdfs/2017PGY_UDAYA-UPreport.pdf

---

## [Editor Report · Decision Letter 2]

14 Apr 2021

Examining the predictors of use of sanitary napkins among adolescent girls: A multi-level approach

PONE-D-20-32727R2

Dear Dr. Patel,

We’re pleased to inform you that your manuscript has been judged scientifically suitable for publication and will be formally accepted for publication once it meets all outstanding technical requirements.

Kind regards,

Srinivas Goli, Ph.D.

Academic Editor

PLOS ONE

Additional Editor Comments (optional):

We are pleased to accept this manuscript in current form.
---

## [Editor Report · Acceptance letter]

23 Apr 2021

PONE-D-20-32727R2 

Examining the predictors of use of sanitary napkins among adolescent girls: A multi-level approach 

Dear Dr. Patel:

I'm pleased to inform you that your manuscript has been deemed suitable for publication in PLOS ONE. Congratulations! Your manuscript is now with our production department. 

Kind regards, 

on behalf of

Dr. Srinivas Goli 

Academic Editor

PLOS ONE